# Can seafloor voltage cables be used to study large-scale circulation? An investigation in the Pacific Ocean.

Jakub Velímský[1], Neesha R. Schnepf[2,3], Manoj C. Nair[2,3], and Natalie P. Thomas[4]

[1]Department of Geophysics, Faculty of Mathematics and Physics, Charles University, Prague, Czech Republic
[2]Cooperative Institute for Research in Environmental Sciences (CIRES), University of Colorado, Boulder, CO, USA
[3]National Centers for Environmental Information, National Oceanic & Atmospheric Administration, Boulder, CO, USA
[4]Department of Atmospheric and Oceanic Science, University of Maryland, College Park, MD, USA

**Correspondence:** Jakub Velímský (jakub.velimsky@mff.cuni.cz)

**Abstract.** Marine electromagnetic (EM) signals largely depend on three factors: flow velocity, Earth's main magnetic field, and seawater's electrical conductivity (which depends on the local temperature and salinity). Because of this, there has been recent interest in using marine EM signals to monitor and study ocean circulation. Our study utilizes voltage data from retired seafloor telecommunication cables in the Pacific Ocean to examine whether such cables could be used to monitor circulation

velocity or transport on large-oceanic scales. We process the cable data to isolate the seasonal and monthly variations, and evaluate the correlation between the processed data and numerical predictions of the electric field induced by an estimate of ocean circulation. We find that the correlation between cable voltage data and numerical predictions strongly depends on both the strength and coherence of the model velocities flowing across the cable, the local EM environment, as well as the length of the cable. The cable within the Kuroshio Current had good correlation between data and predictions, whereas two of the cables

in the Eastern Pacific gyre — a region with both low flow speeds and interfering velocity directions across the cable — did not have any clear correlation between data and predictions. Meanwhile, a third cable also located in the Eastern Pacific gyre had also good correlation between data and predictions — although the cable is very long and the speeds were low, it was located in a region of coherent flow velocity across the cable. While much improvement is needed before utilizing seafloor voltage cables to study and monitor oceanic circulation across wide regions, we believe that with additional work, the answer to our

title's question may eventually be yes.

## 1  Introduction

Evaluating and predicting the ocean state is crucially important for reconciling and mitigating climate change's impact on our planet. Oceanic electromagnetic (EM) signals may be directly related to physical parameters of the ocean state, including flow velocity, temperature, and salinity. This has been known for centuries: in 1832, Michael Faraday was the first to attempt

an experiment of measuring the voltage induced by the brackish water of the Thames River (Faraday, 1832). His study was inconclusive, but since then, marine EM signals have been detected by both ground and satellite measurements (Larsen, 1968; Malin, 1970; Sanford, 1971; Cox et al., 1971; Tyler et al., 2003; Sabaka et al., 2016).

Marine electromagnetic fields are produced because saline ocean water is a conducting fluid with a mean electrical conductivity of $\sigma = 3 - 4$ S m$^{-1}$. As this electrically conductive fluid passes through Earth's main magnetic field ($\mathbf{F} \approx 20 - 70$ $\mu$T), it induces electric fields, electric currents, and secondary magnetic fields. The electric current produced by a specific oceanic flow depends on the flow's velocity, the Earth's main magnetic field, and the seawater electrical conductivity, which in turn depends on salinity and temperature. Thus, ideally, three physical oceanic parameters could be extracted from marine EM studies: velocity, salinity, and temperature. However, extracting multiple parameters would require using multiple oceanic electromagnetic signals (eg. the signals from multiple tidal modes, and perhaps also from circulation) (Irrgang et al., 2017; Schnepf, 2017).

In practice, velocity is the only quantity so far determinable from marine EM data. This was accomplished using a passive seafloor telecommunications cable which recorded the voltage difference between Florida and Grand Bahama Island, a distance of approximately 200 km (Larsen and Sanford, 1985; Spain and Sanford, 1987; Larsen, 1991, 1992; Baringer and Larsen, 2001). As the Florida Current passed over the cable, a voltage was induced and this voltage was directly related to the depth-integrated velocity across the cable (i.e. they determined the transport volume). Since 1985, the National Oceanic and Atmospheric Administration (NOAA) has been using submarine cables to monitor the transport of the Florida Current through the Straight of Florida (Meinen et al., 2020).

While data from seafloor voltage cables have been used to study a variety of geopotential fields (Lanzerotti et al., 1986, 1992a; Chave et al., 1992; Shimizu et al., 1998; Fujii and Utada, 2000; Lanzerotti et al., 2001), NOAA's work in the Strait of Florida is the only case of a seafloor voltage cable being reliable to determine the overlying oceanic flow. Numerical work suggests that cables spanning larger regions should still strongly correlate to the flow velocities (Flosadóttir et al., 1997; Vanyan et al., 1998; Manoj et al., 2010), however, there are many challenges in using longer cables. These challenges are largely due to the myriad of processes which may also induce marine electromagnetic fields, especially across the length of the cable: secular variation (Shimizu et al., 1998), variations in ionospheric tides (Pedatella et al., 2012; Schnepf et al., 2018), geomagnetic storms or longer period ionospheric/magnetospheric signals (Lanzerotti et al., 1992a, 1995, 2001). Additionally, because the cable voltage is produced from the electric field integrated along the entire cable length, the longer the cable is, the more challenging it is to decompose the total contribution to the cross-cable ocean transport in any particular section of the cable.

This study aims to provide a 'first step' answer to the question can seafloor voltage cables be used to study large-scale circulation? To investigate whether it may eventually be feasible to use large-scale voltage cables for monitoring ocean flows, we evaluate the correlation between data from large-scale seafloor voltage cables and numerical predictions of the electric field induced by 3-D ocean circulation velocity fields. While this work builds off of studies using seafloor voltage cables to monitor flow velocity in $\sim$100km wide passages, this study aims to examine this application in basin-wide seafloor voltage cables.

## 2 Data and data processing

This study used hourly data from four seafloor voltage cables (detailed in Table 1): three retired AT&T cables (the HAW cables) and one cable managed by the University of Tokyo's Earthquake Research Institute (the OKI cable). The cables HAW1N and HAW1S are 3805 km long and run parallel to each other from Point Arena, California to Hanauma Bay, Hawaii. As

shown in Figure 1, the parallel cables have very similar data, providing a unique and helpful situation for testing the data processing methods and for comparing the observations to numerical predictions. These three cables have been used in previous studies, including those examining geopotential variations (Chave et al., 1992; Lanzerotti et al., 1992b; Fujii and Utada, 2000), ionospheric phenomena (Lanzerotti et al., 1992a), oceanic tides (Fujii and Utada, 2000), and lithospheric/mantle electrical conductivity (Koyama, 2001).

The first step in processing the hourly data was the removal of geomagnetically noisy days (i.e., days where the geomagnetic Ap index was greater than or equal to 20; see Denig 2015 for more on the Ap index). In this way, we reduce the contribution from magnetic field variations of magnetospheric origin and their induced counterparts. This shrunk the amount of available data by 16.1%-21.6% for each cable. Further reduction of the datasets by using only night-side data is impossible for the HAW cables, spanning multiple time zones, and impractical for the OKI cable due to significant decrease of the dataset size and increase of variance. Next, to remove tidal signals the 12 dominant daily tidal modes were fit to the data via least-squares and then subtracted. The following tidal periods were used: 4 hr ($S_6$), 4.8 hr ($S_5$), 6 hr ($S_4$), 8 hr ($S_3$), 11.967236 hr ($K_2$), 12 hr ($S_2$), 12.421 hr ($M_2$), 12.6583 hr ($N_2$), 23.934472 hr ($K_1$), 24 hr ($S_1$), 24.066 hr ($P_1$), and 25.891 hr ($O_1$). Because the data sets have many gaps exceeding 24 hours in length (for example, see Figure 2), bandpass filtering was not used. The data was then smoothed using cubic splines. For seasonal variations, we used 90 day knots between splines, and for monthly variations, we used 30.5 days between knots. Although the daily variations should directly relate to barotropic wind-forced processes (Irrgang et al., 2016a, b, 2017), because of both the data's hourly time sampling and frequent data gaps, as well as challenges in producing daily numerical predictions, we chose to focus on monthly and seasonal variations. Each step of the data processing is shown in Figure 2. As the final step, the mean value is removed from all time series.

A weakness of this data processing is that it does not prevent inclusion of induced signals due to seasonal changes in ionospheric electromagnetic tidal strength. While we removed tidal signals from a least-squares fit, we applied this fit to the entire dataset and did not attempt to remove seasonal changes in ionospheric tides. Seasonally, ionospheric tides can significantly change amplitude (Pedatella et al., 2012), and the horizontal components of these tides are likely to induce signals at the ground (Schnepf et al., 2018), however, attempting to constrain seasonal changes in tidal strength is challenging. Ideally, the least-squares fit could be conducted on shorter intervals of the data, but this worsens the accuracy of the least-squares inversion. Ionospheric field models could be used, but this would also introduce unknown error quantities. Thus, we did not attempt to remove seasonal changes in tidal amplitude but remind the reader that these signals may influence the monthly and seasonal variations. The contribution of the main field secular variation is not removed from the data as it is included in the numerical calculations described in the next section.

## 3 Numerical predictions of ocean circulation's electric field

We numerically predict the electromagnetic signals produced by ocean circulation using the time-domain numerical solver elmgTD of the electromagnetic induction equation (Velímský and Martinec, 2005; Velímský, 2013; Šachl et al., 2019; Velímský

et al., 2019),

$$\mu_0 \frac{\partial \mathbf{B}}{\partial t} + \nabla \times (\frac{1}{\sigma} \nabla \times \mathbf{B}) = \mu_0 \nabla \times (\mathbf{u} \times \mathbf{F}). \tag{1}$$

Here $\mathbf{B}(\mathbf{r};t)$ is the induced magnetic field, $\mathbf{u}(\mathbf{r};t)$ is the velocity, $\mu_0$ is the magnetic permeability of vacuum, $\sigma(\mathbf{r};t)$ is the electrical conductivity, and $\mathbf{F}(\mathbf{r};t)$ is the main geomagnetic field. The observable electric field $\mathbf{E}(\mathbf{r};t)$ is obtained from the induced magnetic field by post-processing,

$$\mathbf{E} = \frac{1}{\mu_0 \sigma} (\nabla \times \mathbf{B}) - \mathbf{u} \times \mathbf{F}. \tag{2}$$

The elmgTD time-domain solver is based on spherical harmonic parameterization in lateral coordinates, and uses 1-D finite elements for radial discretization. The model is fully three-dimensional, incorporating also the vertical stratification of the ocean electrical conductivity and of the velocities, and accounting for the effect of variable bathymetry. Moreover, the seasonal variations of the ocean electrical conductivity, and the secular variations of the main field are taken into account. The solution includes both the poloidal and toroidal components of the induced magnetic field (Šachl et al., 2019; Velímský et al., 2019), thus allowing for the inductive and galvanic coupling between the oceans and the mantle, as well as self-induction within the oceans. Numerically, the linear system is solved by the preconditioned iterative BiCGStab(2) scheme (Sleijpen and Fokkema, 1993) with massive parallelization applied across the time levels.

Monthly values of the horizontal and vertical components of ocean velocity from the data-assimilated model Estimating the Circulation and Climate of the Ocean (ECCOv4r4) (Forget et al., 2015; Fukumori et al., 2017) were input into the elmgTD solver to compute the electromagnetic fields they induce from January 1997 to November 2001. Along with the monthly velocity values from ECCO, monthly values from the International Geomagnetic Reference Field (IGRF) (Finlay et al., 2010) were used for the main field, and monthly climatological data from NOAA's World Ocean Atlas (WOA) were used to describe the global seawater electrical conductivity $\sigma$ (Tyler et al., 2017). The conductivity model also includes the coastal and ocean sediments on the seafloor with thickness distribution and conductivity values following Everett et al. (2003).

Figure 3 illustrates these inputs used for the elmgTD numerical solver. The vertical velocity is not shown here. Although it is included in our calculations, as it represents only a minimum additional computational burden, its effect on the induced fields is negligible. Underlying these inputs, the electrical conductivity of the mantle follows the 1-D global profile obtained by inversion of satellite data (Grayver et al., 2017).

In the present calculations, we truncate the spherical harmonic expansion at degree 240, corresponding to approximately $0.75 \times 0.75$ degree resolution. The radial parameterization within the oceans uses 50 shell layers, following the irregular discretization of the ECCO model. The seawater monthly conductivities from NOAA's WOA were interpolated to the same grid via bilinear formula in angular coordinates, and weighted, conductance-preserving averaging in radius.

The model was run from January 1997 through the end of November 2001. Global results were extracted from the middle of every month (e.g., 1997-01-17, 1997-02-15, 1997-03-18, 1997-04-17, etc.), but daily results were extracted along the transect of the cables' paths.

To compare numerical predictions with the processed seafloor cable observations, the electric field was integrated along the seafloor between the endpoints of each cable. For each cable element, the electric field component along the cable direction was

calculated in the lowermost ocean discretization layer. The linear trend was finally removed from each time series of predicted cable voltages.

## 4    Results and discussion

Figures 4, 5, and 6 summarize the processed voltages, and their numerical predictions from the elmgTD ECCO-based simulation for individual cables. The top panel in each figure shows the time series of cable voltages processed with the 90-day knotted spline fit and the 30.5-day knotted spline fit in red and green, respecively. In the case of the HAW1 cables, the N and S branches are distinguished by solid and dashed lines. The blue line then shows the results of the numerical predicitions. A linear trend was removed from all shown time series. The middle panels in Figures 4, 5, and 6 show the numerical predictions

of the voltage gradient (i.e., the electric field) on the seafloor, along the respective cables, before integration. Finally, the bottom panels display the transport $T_\perp$ of the ECCO model across each cable. Note that by transport in this context we denote the vertically integrated velocity component perpendicular to the cable for each cable element position and time, hence the unit of $\mathrm{m}^2/\mathrm{s}$. Although it is not a direct input to the numerical simulations (contrary to the velocities in individual ECCO layers), it serves as a useful proxy for discussions below.

Looking first at the common features of the results for all cables, we note, as expected from basic geometrical considerations, a general similarity between the voltage gradient along the cable, and the water transport across the cable $T_\perp$. We can use these to discuss the effect of individual currents on the numerical predictions. However, while the ocean flows are certainly the dominant term controlling the induced electric fields, the additional contributions of other effects yield much richer spatio-temporal structure. The main field variations in both space and time can have a linear impact on the large-scale features, as

implied by the forcing term of the EM induction equation (1). Moreover, the local variations of seawater conductivity, the bathymetry, and the sediment thickness affect the electric field in a non-linear way. In particular, the toroidal magnetic mode, which corresponds to the poloidal electric currents, and which stems from the galvanic coupling between the ocean and the underlying solid Earth, can play an important role (Chave et al., 1989; Velímský et al., 2019).

In closer inspection of the OKI cable results, the importance of the Kuroshio current stands out, at the distance of 300–600

145    km from Honshu (Figure 4, bottom panel). It produces by far the largest contribution to the predicted voltages (middle and top panels). In terms of spatial distribution, the positions along the cable where the largest contributions to the electric field are induced do not match with the peak positions of the cross-cable transport. This discrepancy can be attributed to the electrically strongly heterogeneous environment caused by large bathymetry changes in the vicinity of the Ryukyu arc. The ECCO model suggests an increase of the transport in the last months of 2000, which is consequently responsible for the increased voltage

in the numerical model. However, no such increase is present in the observed voltages, and this discrepancy remains an open question. If we trust the OKI voltages, it is possible that the ECCO model is overestimating the Kuroshio strength by the end of 2000.

In the case of HAW1N and HAW1S, the numerical model predicts significantly smaller amplitudes of cable voltage variations when compared to the observations (Figure 5). The California current is the main contributor to the total voltages, at distances

up to 1000 km from the Californian coast. The spatio-temporal distribution of the cross-cable transport and the induced voltages is in good agreement due to an uncomplicated electrical conductivity distribution in the deep ocean. The ocean transports across the HAW1 cables demonstrate larger seasonal variations than in the case of Kuroshio. However, lack of significant contributions perpendicular to the cable, and changing direction of these flows both along the cable, and in time, yield poor agreement of the total integrated voltage with the observations.

The HAW3 cable, on the other hand, shows good agreement between the predicted and observed voltages (Figure 6). The numerical model is again dominated by the California current, which is here closer to the coast. The HAW3 cable lies a bit to the south of the HAW1N&S cables and it is also within the low speed region of the Eastern Pacific Gyre. The transport across the cable in the central Pacific is more coherent, yielding slightly stronger signals than in the case of the HAW1 cables. Again, the pattern of the cross-cable transport is well matched with the spatio-temporal map of the induced voltages.

In Table 2, we calculated two sets of correlation coefficients. In the second column of the table, the voltages predicted by the numerical model were correlated with the total ECCO-based water transport (in $\mathrm{m^3/s}$) across the respective cables,

$$P_\perp = \int\limits_{\mathrm{start}}^{\mathrm{end}} T_\perp \, \mathrm{d}l. \tag{3}$$

These values are independent on the actual cable voltage measurements, and can provide an upper limit on what can be achieved by interpretation of long-cable voltages in terms of ocean flows. Large correlations were obtained for HAW1 and HAW3 cable locations, while the integrated flow across the OKI cable is poorly correlated with the predicted voltage. This stresses the importance of accurate modelling of the induced electric field in strongly heterogeneous areas.

In the third column of Table 2 we show the correlation coefficients between the predicted and observed voltages using the 30.5-day knot separation datasets. Because of gaps present in the data, the Gaussian-kernel method (Rehfeld et al., 2011) was applied. It is obvious that the discrepancies between the predicted and observed voltages are still large, and significant efforts are required both on the side of data processing and numerical modeling to reconcile the results. The OKI cable in particular presents an interesting case. Although the total cross-cable transport is less correlated with the predicted voltages than in the case of both HAW1 cables, the agreement with the observations is considerably better. This again points to the role of local EM effects.

On the side of numerical modelling, one could devise a comparison study between different ocean models. Indeed, we have used our model to predict the magnetic fields of the LSOMG model in the past (Velímský et al., 2019), and we have also attempted the calculation of the cable voltages for the eddy-resolving GLORYS ocean model (not shown here). One problem related to this approach is the volume of computational resources necessary to carry out the calculations. As the cable voltages are sensitive to local electric fields, the usual simplifications of the EM induction solver, based on the thin-sheet approximation, or representing the oceans by a single layer with integrated water transports and electrical conductances, are problematic (Šachl et al., 2019; Velímský et al., 2019). The single 5-year calculation of the full physical model presented here, with 50 ocean layers and spherical-harmonic truncation degree 240, required about $10^5$ CPU-hours to complete. Semi-global or regional modelling tools with local refinement ability are needed for more accurate numerical studies.

The qualitative comparison of the induced voltages and water transports along the cables, as presented in this paper, could be made more exact by applying the Principal Component Analysis/ Empirical Orthogonal Functions methodology. When applied only to the water transports provided by different ocean models, it could reduce the burden of calculating a detailed 3-D EM response to each model, and allow a more focused interpretation of the observed voltages. We plan to carry out such analysis in the future.

The studies by Larsen (1992) evaluating transport in the Straight of Florida from seafloor voltage cable data had correlation values corresponding to much higher values than those of this study. As shown in Figure 20 of his paper, his correlation squared values ranged from 0.61 to 0.94. However, Larsen's study was fundamentally different: the seafloor voltage cable was an order of magnitude shorter than the cables considered in this study and the Gulf Stream within the Strait of Florida has large speeds, as well as coherent velocities flowing perpendicularly to the cables, so Larsen's study overall had a more ideal signal-to-noise ratio.

## 5   Conclusions

We present an evaluation of using seafloor voltage cables for monitoring circulation across oceanic basins. We compare processed seafloor voltage cable data with the numerical predictions produced using an electromagnetic induction solver, fed by flow velocity estimates from the data assimilated model ECCO and seawater electrical conductivity climatologies from the NOAA World Ocean Atlas. We find that the correlation between cable voltage data and numerical predictions strongly depends on both the amplitude and direction of the flow velocities across the cable.

Due to the computational constraints, the calculations of the ocean-induced electric field presented here are limited to a single realization of ocean velocity estimates: the ECCO model. Therefore, beside the unmodelled or uncorrected signals in the seafloor cable voltages, a first-order source of discrepancy between the numerical prediction and observation is the inaccuracy of the flow velocity estimates. An extended analysis and comparison of the cable voltages calculations driven by other ocean circulation models is therefore desirable in the future as well as taking into account direct velocity observations (Szuts et al., 2019).

While much improvement is needed before utilizing seafloor voltage cables to study and monitor ocean circulation across large regions, we believe that seafloor voltage cables can eventually be used to study and monitor large-scale ocean flow. The cables used in this study were installed for telecommunication purposes — there was no regard for whether these cables would be best suited to monitor ocean currents. Flow information can most reliably be extracted from seafloor voltage cable data when the flow has mostly unidirectional, perpendicular velocities across the cable. For our study, the OKI cable was in the area with the largest velocities, but because it is oriented mostly parallel to the Kuroshio Current, its correlation would likely greatly improve if it was instead perpendicular to the current's flow.

If voltage cables were strategically placed on the seafloor between Antarctica and Chile (a distance of $\sim 700$ km), or Antarctica and New Zealand (a distance of $\sim 1300$ km), because of both the shorter cable length (as compared to the HAW1 and HAW3 cables) and the relatively uniform and large flow velocities, the correlation between data and predictions could be quite

high. Indeed, seafloor voltage cables may be a very effective method for measuring and continuously monitoring the flow of the Antarctic Circumpolar Current — this is definitely something worth investigating.

Using existing cables, the correlation between data and numerical predictions will likely also improve if methodology is enhanced to remove induced signals from seasonal variations in ionospheric signals.

*Data availability.* The data and numerical predictions discussed in this study are freely available for download at the website geomag.colorado.edu/OCEM.

*Author contributions.* N. R. S. and M. N. conceived the questions and methodology of this study, and wrote the initial version of the manuscript. N. R. S. and N. P. T. worked on the processing of cable data. M. N. supervised N. R. S. and N. P. T. on work related to this project and provided useful feedback on improving the manuscript. J. V. carried out the numerical modelling, and also contributed to the 230 manuscript, in particular the revised version.

*Competing interests.* No competing interests are present.

*Acknowledgements.* N. R. S. was supported by NASA grant 80NSSC17K0450. N. R. S. and M. N. were also supported by a CIRES IRP grant. J.V. acknowledges the support of the Grant Agency of the Czech Republic, project No. P210/17-03689S. The computational resources were provided by The Ministry of Education, Youth and Sports, Czech Republic, from the Large Infrastructures for Research, Experimental 235 Development and Innovations project "IT4Innovations National Supercomputing Center - LM2015070", project ID OPEN-13-21. We thank two anonymous reviewers for their helpful comments.

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

**Table 1.** The seafloor voltage cables used in this study. The HAW1N and S cables run parallel to each other.

| Cable | Starting location | Ending location | Length (km) | Timespan |
|---|---|---|---|---|
| HAW1N&S | Point Arena, CA, USA | Hanauma Bay, HI, USA | 3805 | 04/1990–12/2001 |
| HAW3 | San Luis Obispo, CA, USA | Makaha, HI, USA | 3946 | 08/1994–07/2000 |
| OKI | Ninomiya, Honshu, Japan | Okinawa, Japan | 1447 | 04/1999–12/2001 |

**Table 2.** For individual seafloor cables, we show in the second column the correlation coefficients between the cable voltages predicted by the numerical model, and the total ECCO-derived water transport across the cable $P_\perp$. The third column shows the correlation coefficients between the predicted and observed voltages for 30.5-day spline knot separation.

| Cable | $\mathbf{corr}(U^{\mathrm{pred}}, P_\perp)$ | $\mathbf{corr}(U^{\mathrm{pred}}, U^{\mathrm{obs}})$ |
|---|---|---|
| HAW1N | 0.92 | 0.28 |
| HAW1S | 0.92 | 0.11 |
| HAW3 | 0.78 | 0.51 |
| OKI | 0.33 | 0.42 |

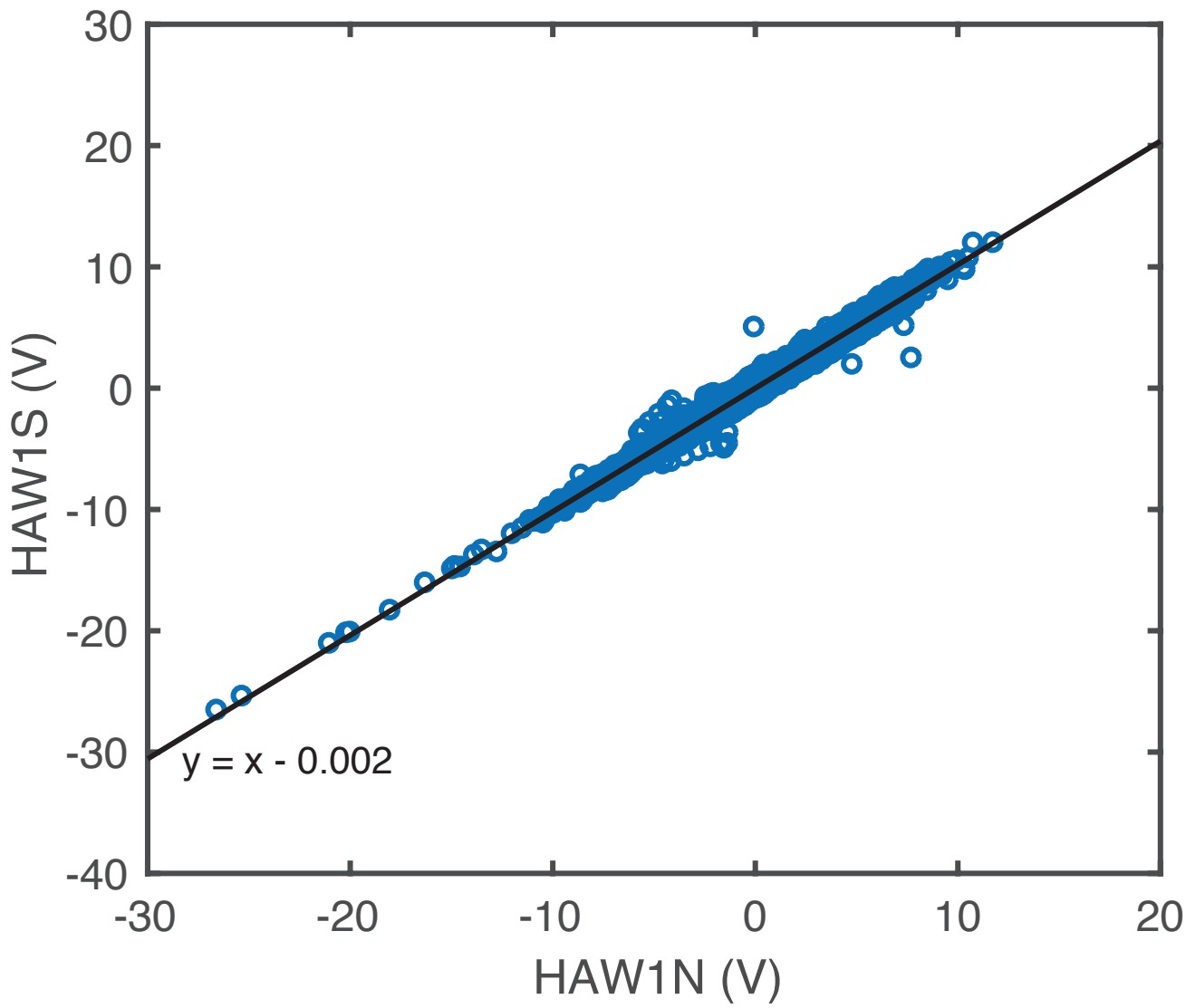

**Figure 1.** The voltage data of HAW1N versus HAW1S is shown in a correlation scatter plot. As shown by the line of best fit ($y = x - 0.002$), the data from the two cables match very closely.

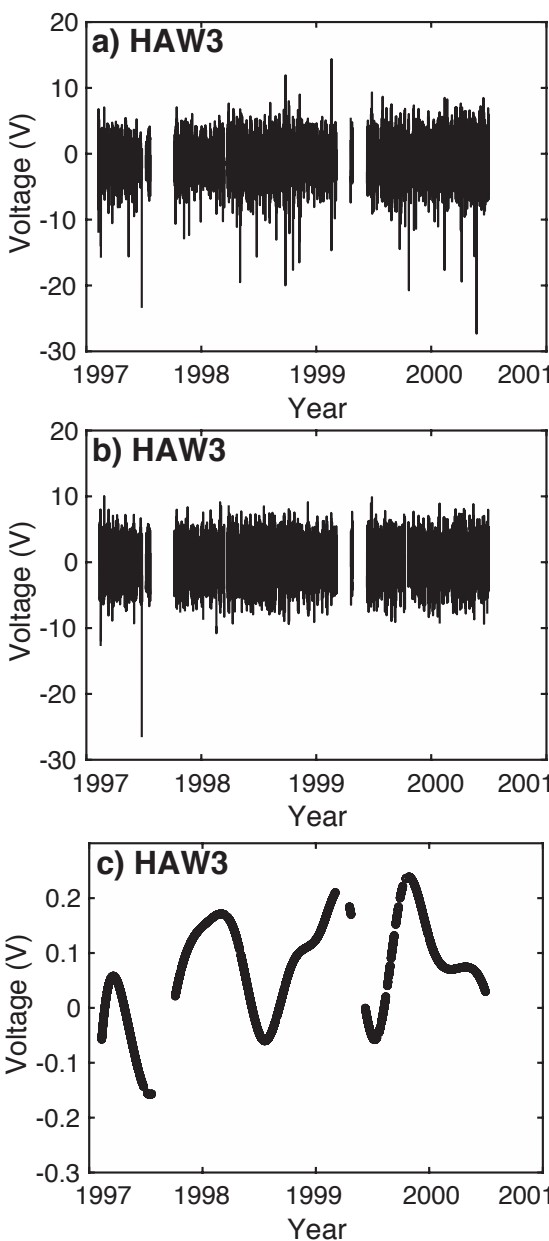

**Figure 2.** Each step of the data processing is shown here using HAW3 as an example: a) the raw time series, b) the time series with days of Ap > 20 removed and tidal signals also removed, and c) the smoothed time series produced by splines with 90 day knots.

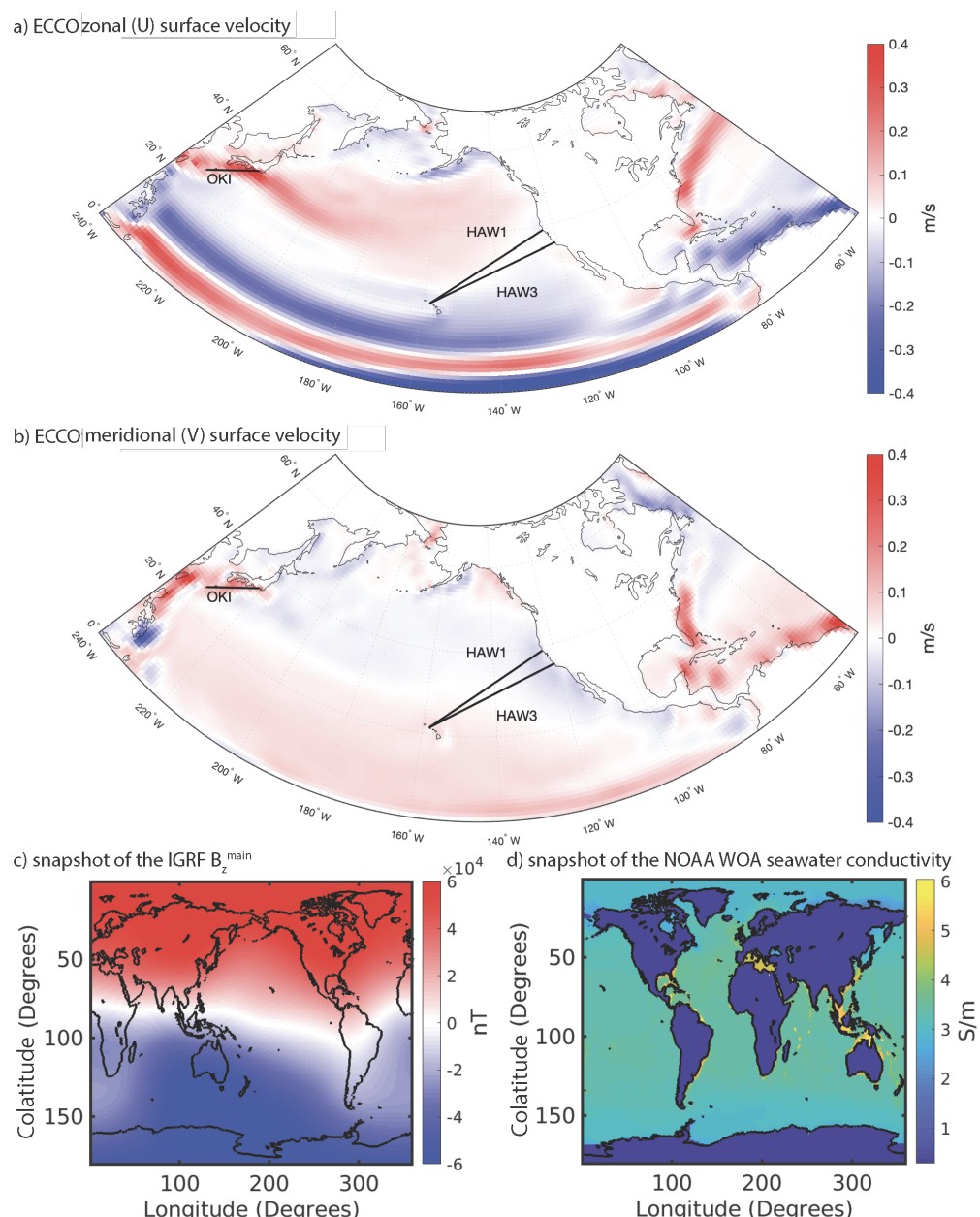

**Figure 3.** The surface velocities from ECCO are shown in a) for the zonal (U) component and b) for the meridional (V) component. The labelled, thick black lines denote the seafloor voltage cables used in this study. A snapshot of the IGRF vertical main field, $B_z^{main}$, from January 17, 1997 is illustrated in c) and d) depicts the NOAA World Ocean Atlas seawater electrical conductivity's January climatology in the surface layer.

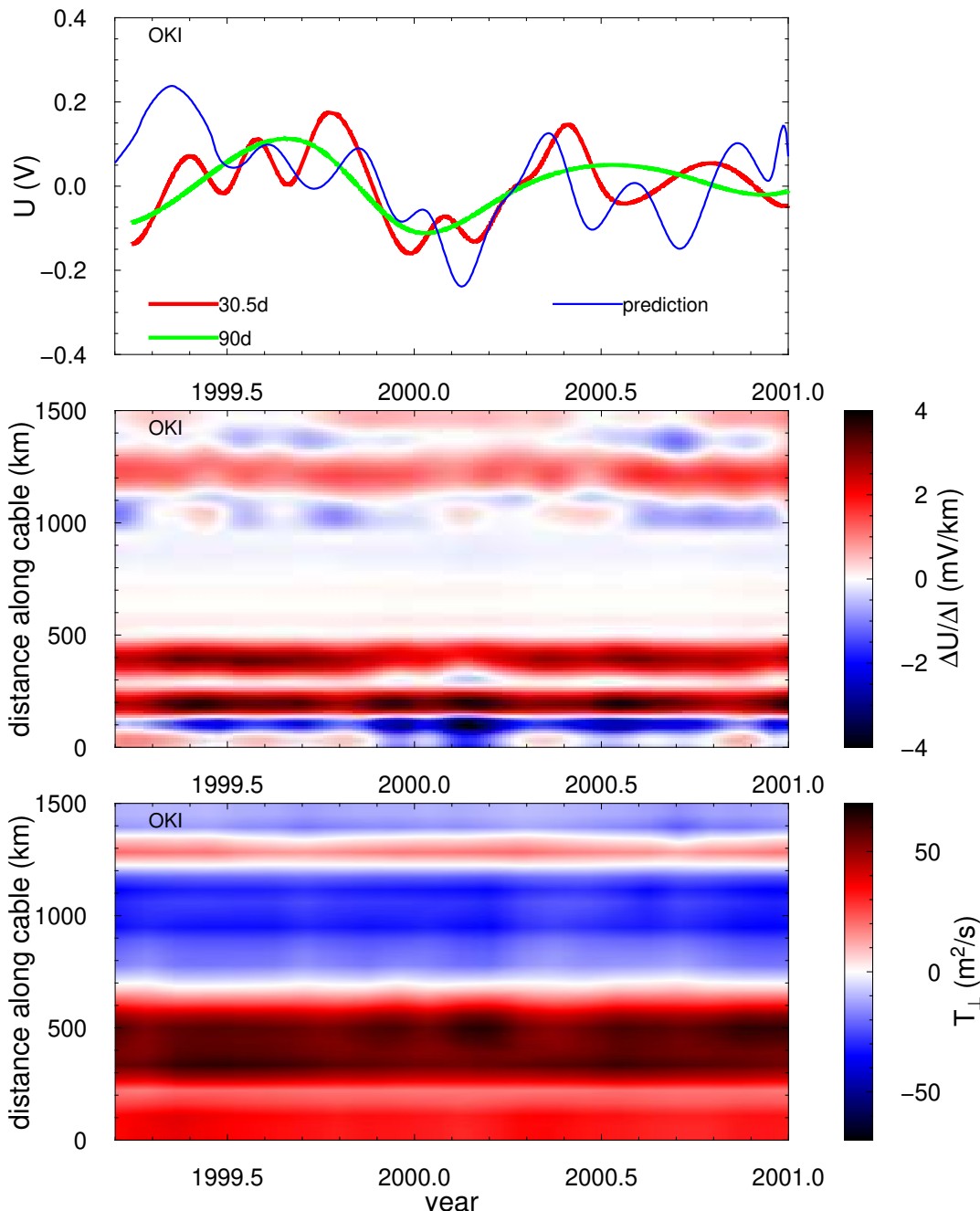

**Figure 4.** The results for the OKI cable. The top panel shows in red and green the smoothed time series of cable voltages using 30.5-day and 90-day knot separation, respectively. The blue line correspond to the predictions obtained by the numerical model. The middle panel shows the time-development of voltage gradient along the cable length from the 3-D model. In the bottom panel, we plot in similar way the ECCOv4r4 vertically integrated transport across the cable. The cable orientation follows Table 1, from Honshu to Okinawa.

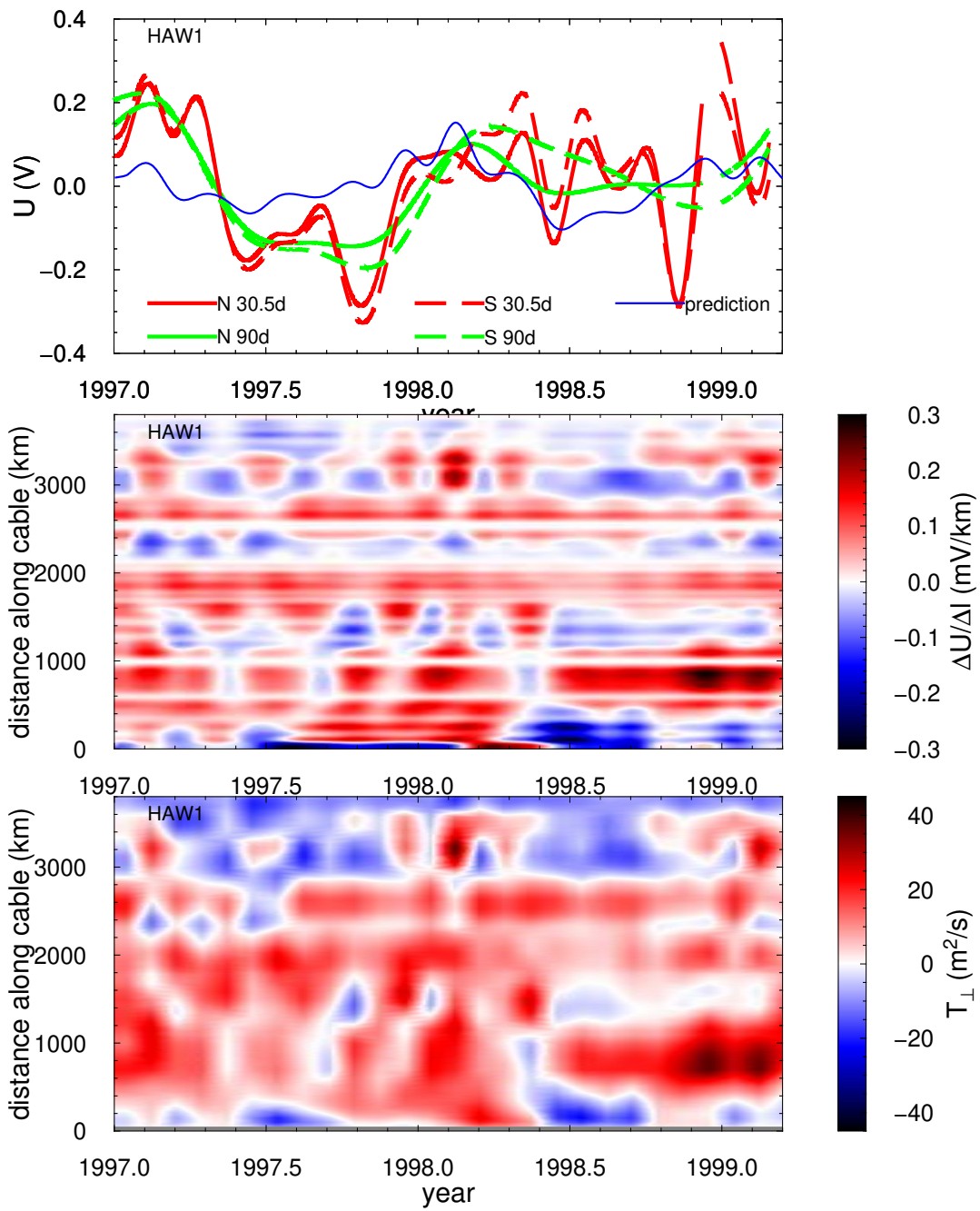

**Figure 5.** The results for the HAW1 cables. The N and S cables are distinguished by solid and dashed lines in the top panel. The cable orientation is from California to Hawaii. Otherwise, the description corresponds to Figure 4.

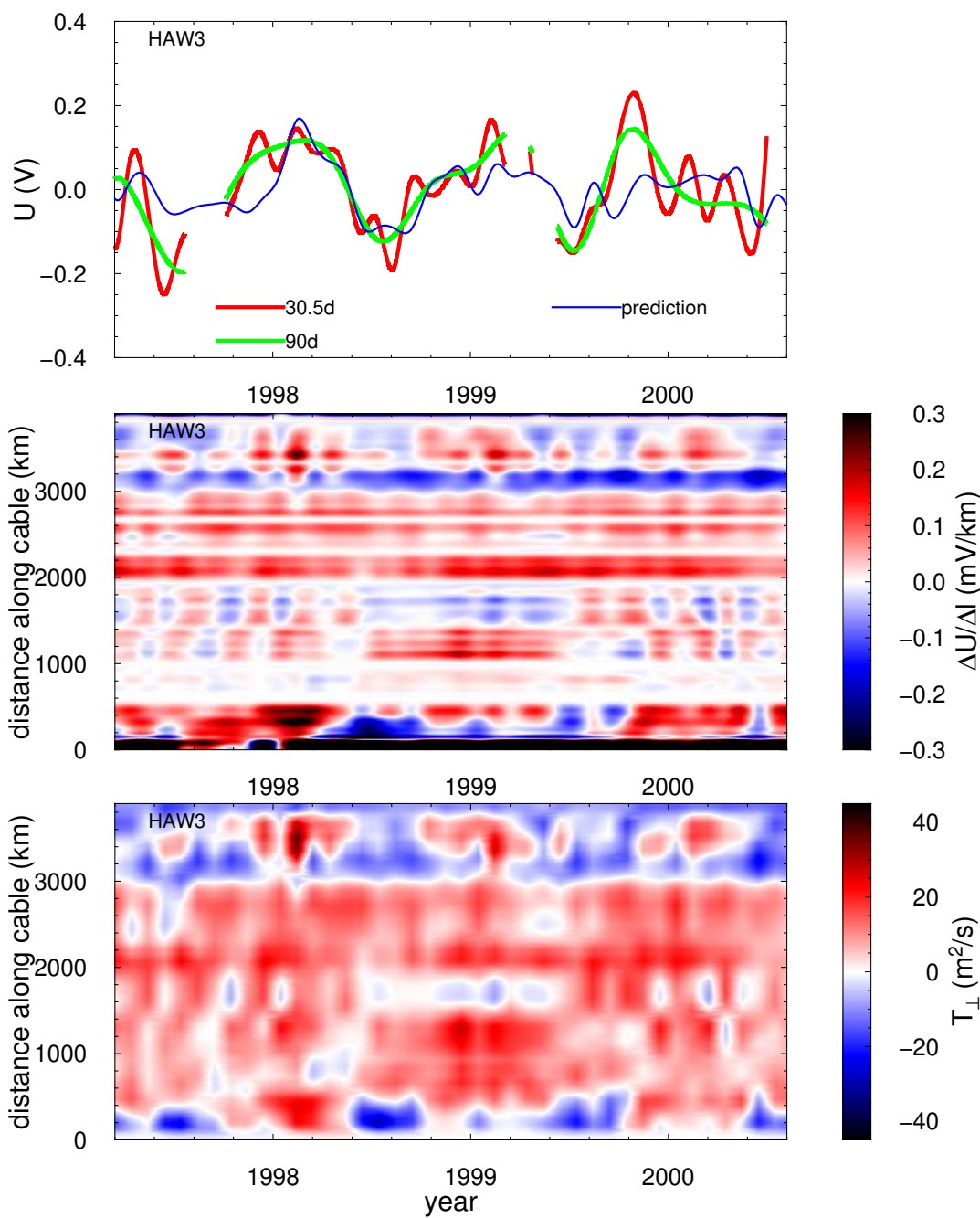

**Figure 6.** The results for the HAW3 cable. The cable orientation is from California to Hawaii. The description corresponds to Figure 4.