# Peer review of "Can seafloor voltage cables be used to study large-scale circulation? An investigation in the Pacific Ocean."

_Ocean Science, 2019_

## Referee Comment (RC1) · Anonymous Referee #1 · 10 Feb 2020

The paper reports on the use of deep sea telecommunication cables for ocean monitoring. The paper is well ordered, well worded and easy to follow. The topic itself is highly relevant, given the lack of data for the ocean and the major part of oceanic processes in the (changing) climate system. Overall the paper could do more to motivate its relevance and to separate itself from similar previous studies. I see flaws in the papers conclusions or at least in the discussion of the conclusions.

Scientific remarks:

1 - Introduction: The introduction is good and short. However, the relevance of the study could be stressed more. It should be a stated clearly that the paper focuses on

oceanic velocities only. The differences to the well cited previous studies (e.g., Larsen et al.) should be made very clear.

2 - Data and processing: The applied methods seem a bit arbitrary. I would recommend using band-pass filters in stead of splines. The splines' impact on the spectrum is not straight forward.

Daily (and sub-daily?) tides are removed before the spline smoothing. Why is this necessary? Again, better use a band pass filter.

Some signals are not removed or discussed: trends, secular variation (accounted for in sec.3), solar cycle, ionospheric and magnetospheric effects. Some of these are mentioned in the introduction but should in addition be discussed here. The ionosphere is only mentioned with a seasonal influence on the tides. But surely it can have a direct seasonal influence? As an alternative the study could focus on night side data alone.

Longer signals are not removed (band pass filter?). It would be a good idea to at least remove the trends before calculating correlations with ECCO. ECCO may have very different trends for different reasons. Furthermore, the use of climatological conductivity in elmgTD may falsify the trends of the ECCO EM results.

3 - EM Prediction: Why are only the horizontal ECCO velocities used? Can the influence of the vertical velocities and their changes be quantified?

To use climatological conductivity should at least discussed in sec. 4. It may be better to use the conductivity from ECCO. Or booth, to demonstrate the effect of sigma on the results.

Are sediments considered in elmgTD?

Line 107: The elmgTD was forced with monthly velocities, monthly core fields, monthly conductivity but produces daily estimates? If so, do they contain any information on shorter than monthly time scales?

Why are the ECCO data not spline smoothed as well?

Line 110: What is meant by layer most closely corresponding to the sea floor The models bottom layer? Do elmgTD and ECCO use a different bathymetry or does elmgTD bathymetry does not fit well to the reality? Maybe I just don't understand this sentence.

4 - Results and discussion: In my opinion, some conclusions lack a solid base. To amend this I would strongly recommend some recalculations or additional analyses. At the very least, the discussion should be deepened. The sections main arguments base on a mismatch between the ECCO results and the cable estimates. In short, two very different data sets are compared and if the observations do not fit to the model based estimates, then the observations are said to have a low "signal-to-noise-ratio".

The ECCO based results are not questioned or discussed at all

I have several questions here: How reliable are the ECCO results? What are the errors of the used velocities? There probably exist model inter comparison studies. I would advise to repeat the elmgTD calculations with at least two other ocean models.

What is the influence of the used climatological conductivity? It would be a good idea to recalculate the ECCO based results with ECCO based conductivity which shows real intra-annual variability and discuss the differences.

Is the quality of the modeled data globally uniform? If not (probably not), then the "signal-to-noise-ration" mentioned in the paper depends not on cable length or the strength and uniformity of ocean currents but may just depend on location. ECCO is an assimilative ocean model: that means, that the errors of the results depend on the amount and quality of available data. This is not globally uniform, too. For example, if satellite altimetry is assimilated (major source of ECCO's information), then a big current like the Kuroshio (OKI cable) has a strong sea surface gradient and will be much better represented by the assimilated model than the more or less "flat" oceanic areas (HAW). But the HAW measurements might still be not worse than the OKI measurements. As long as it is not clear if the found low correlations are caused by the model or the observations or the principal differences in the data, one should not call them low signal-to-noise ratio.

The differences between modeled data and observed data are not discussed enough Please discuss the effects in the cable data that are not in the modeled data: trends, ionosphere, solar cycle etc. see remarks to Sec. 2

Please discuss the representation error/issue: Grid box averages are compared to a very local cable path. By looking at Fig. 5C, one can see that even very similar cable paths can already produce very different results. Please discuss this. In addition, from the differences between HAW1N and HAW1S some real signal-to-noise ratio could be derived. An error bar probably could be produced that sets the model to observation comparison into relation. Is there any explanation, why these two cables produce different time series (surely, the effects mentioned in the previous paragraph should affect both cables equally)?

Again, for the model side an error bar should be generated or estimated, too.

Spline smoothed observations are compared to temporal averages from the model?

---

## Referee Comment (RC2) · Anonymous Referee #2 · 25 Feb 2020

The authors go through the commendable and accurate process of estimating oceanic electric fields from models, to compare with data from 4 submarine cables. The primary results presented are correlations between the observed and modelled electric fields, which are used to infer the suitability of using submarine cables for oceanic velocity. The statistical interpretation of these correlations does not seem methodical enough to be believable in its current state. The conclusions presented are not detailed, and do not advance the field beyond earlier papers on the topic. Even their recommendations for placing cables in strategic points - an easy thing to propose but much harder to actually implement, see the SMART cable effort - does not include the specificity needed to ensure that such cables can provide useful results for inferring ocean circulation, such as resolving meanders, variables subsurface sediment thickness, or flow acceleration/deceleration. This article focuses on just the first step of getting useful cable voltage measurements, obtaining a high correlation between observations and models, but the second step of interpreting why the cable voltages change is just as important and even harder.

Technical comments:

Intro

lines 41-43: Another confouncing factor is that, because longer cables integrate over longer distances, it becomes harder to assign transport or velocity to any single section of the cable.

lines 44-45: This question has already been addressed in the literature.

Data and Data processing

line 65: Also look at Luther publications from BEMPEX for an interpretation of the oceanic EF response at periods from hours to days.

Section 3

lines 86-93: Does elmgTD also include mildly conductive subsurface sediment layers, which varying significantly across ocean basins? Theses are important for interpreting oceanic EM signals.

Figure 3: What date/time are the ECCO velocities shown for?

Figure 4: Why is this shown globally, when the focus is on the North Pacific? It would be more instructive to show the signed electric field across the cables, as the fact of taking the absolute magnitude hides the important fact of current reversals across the cables? What time point do these figures apply to? The two pairs of plots (a and b; c and d) are slightly redundant: it doesn't matter what the surface or bottom fields/forcing are, but rather what their depth-integrated quantities are. Suggest reducing this figure

to 2 plots showing the depth-averaged quantities.

Section 4, Results and Discussion

line 119: I am highly surprised that "all of the p-values were equal to 0". Given that a p-value is positive definite by definition, I expect this to be impossible, certainly with real-world noisy data. This statistical significance testing needs to be redone more accurately. You need to also account for the degrees of freedom of a low-frequency signal.

For interpreting the HAW1N, HAW1S, and HAW3 voltages, it would be instructive to do statistical tests (see earlier comments) to see if the correlations of these cables with the model data are statistically distinguishable from each other. I am doubtful that they are.

lines 153-156: This is the crux of successfully using submarine cable voltages: placing it in a region that is conducive to interpreting such measurements. Note also that substantial effort is put into calibrating the Florida Current voltage time-series, see more recent publications by Meinen.

lines 157-163: Yes, most scientists who work with submarine cables could confirm that these are useful requirements for using such signals to interpret voltages. This point is not, however, substantiated in detail by this paper.

Nowhere do the authors note that their correlations are subject to an important additional source of noise: that the ECCO model might not accurately reflect the actual monthly averaged oceanic velocity field. To my knowledge nobody is able to evaluate ocean models based on their velocity field (for many practical and technical reasons). In light of this, a better approach, see Flosadottir et al 1997, would be to use a "perfect model" approach, so that you don't have to worry about the mismatch between ocean models and actual ocean circulation.

Also, for understanding the Florida Cable results, important details are presented in
Spain and Sanford, J Mar Research, 1987.

---

## Referee Comment (RC3) · Anonymous Referee #2 · 25 Feb 2020

To go along with my review above, what would actually be a very good contribution to understanding these 4 cables would be to delve into the oceanic flow that causes the voltages. Within the ECCO "perfect model" framework, how does variability and spatial structure of the oceanic flow field give rise to the cable voltages? This would provide a way to quantify the paper's unsubstantiated comments about the relative impact of coherent or incoherent flow. As cable voltage depends on the water transport above it, you could break the cables voltage calculation from the ECCO model into shorter segments, and compare the multivariate segment-induced voltage against the transport over the segment. An EOF or Principal Component analysis seems best for explaining the voltages, and then coherence between the different segments can be

calculated to quantify the degree of coherent cross-cable flow or not. Kudos if you look at variable segment lengths, especially calling out the spacing typical of telecom repeaters (see SMART cable material).

---

## Author Comment (AC1) · 24 Jun 2020

The paper reports on the use of deep sea telecommunication cables for ocean monitoring. The paper is well ordered, well worded and easy to follow. The topic itself is highly relevant, given the lack of data for the ocean and the major part of oceanic processes in the (changing) climate system. Overall the paper could do more to motivate its relevance and to separate itself from similar previous studies. I see flaws in the papers conclusions or at least in the discussion of the conclusions.

Scientific remarks:
1 - Introduction: The introduction is good and short. However, the relevance of the study could be stressed more. It should be stated clearly that the paper focuses on oceanic velocities only. The differences to the well cited previous studies (e.g., Larsen et al.) should be made very clear.

Thank you for this comment. The final paragraph of the introduction will be accordingly revised to be explicit about these differences:

> This study aims to provide a 'first step' answer to the question can seafloor voltage cables be used to study large-scale circulation? To investigate whether it may eventually be feasible to use large-scale voltage cables for monitoring ocean flows, we evaluate the correlation between data from large-scale seafloor voltage cables and numerical predictions of the electric field induced by 3-D ocean circulation velocity fields. While this work builds off of studies using seafloor voltage cables to monitor flow velocity in ~100km wide passages, this study is the first to examine this application in basin-wide seafloor voltage cables.

2 - Data and processing: The applied methods seem a bit arbitrary. I would recommend using band-pass filters instead of splines. The splines' impact on the spectrum is not straight forward.

Unfortunately, while I do agree that a band-pass filter would be more straight-forward (and perhaps less arbitrary) than using splines, because 1) the data is not continuous and 2) data gaps are often long enough that interpolating would cause more problems than it would solve, we avoid using frequency/spectral based methods. This is why we instead use splines.

Daily (and sub-daily?) tides are removed before the spline smoothing. Why is this necessary? Again, better use a band pass filter.

Because we cannot use a bandpass filter, we remove tidal signals before the spline smoothing to ensure that they are not influencing the spline fit.

We will explicitly state in our paper that we remove tides of the following periods: 4 hr, 4.8 hr, 6 hr, 8 hr, 11.967236 hr, 12 hr, 12.421 hr, 12.6583 hr, 23.934472 hr, 24 hr, 24.066 hr, and 25.891 hr.

Some signals are not removed or discussed: trends, secular variation (accounted for in sec.3), solar cycle, ionospheric and magnetospheric effects. Some of these are mentioned in the introduction but should in addition be discussed here. The ionosphere is only mentioned with a seasonal influence on the tides. But surely it can have a direct seasonal influence?

We will be sure that the methods section will be revised to clearly state what signals are removed or left as a source of error. We do remove the direct current (DC) trend in the data and most magnetospheric signals are also removed by limiting the study to quiet times (Ap < 2 nT). Because we cannot separate

secular variation in the cable data, we incorporate it into the numerical simulations by using time-dependent IGRF model values in the forcing term (and note that the IGRF model also does not separate the internal signal induced by secular variation).

It is certainly possible that there is a direct seasonal influence from the ionosphere since ionospheric signals depends on sunlight and that varies seasonally. Indeed, there may be noise from other sources (eg. seasonal changes in the quiet magnetosphere) that is not removed by the methods we have undertaken. This would be the case even if we used bandpass methods: noise within the same frequency band as ocean circulation would still leak into our study.

As an alternative the study could focus on night side data alone.

This is only possible for one of the cable's in our study: the OKI cable. The other cables span across too many time zones, so limiting the data to times when it is night across the entire cable limits the data too much. Because the OKI cable mostly varies with latitude rather than longitude, we were able to utilize night-time only data and perform our analysis on that data. For this analysis, local night-time was determined as the time between local sunset and local sunrise at the mean latitude and mean longitude coordinate for the OKI cable.

Comparing OKI's nighttime data to our numerical simulations is not ideal because by only using night-time data, we are decreasing the sample size of our dataset and increasing the variance (the variance of all OKI data is 1.8798 V, whereas for night-only data it is 2.1250 V).

| Cable | 30.5 spline fit | | 90 spline fit | |
|---|---|---|---|---|
| | R | $R^2$ | R | $R^2$ |
| OKI | 0.5920 | 0.5920 | 0.6526 | 0.4259 |
| OKI night-only | -0.1437 | -3.9602 | -0.1819 | -7.4372 |

[Figure]

Left: plot of the OKI cable's nighttime processed data using a spline fit of 30.5 day knots.

[Figure]

Left: plot of the OKI cable's nighttime processed data using a spline fit of 90 day knots.

As shown in in the above figures and table, the data and numerical simulations now have voltages of more similar magnitude. However, the simulation still has a much narrower range of voltage compared to the processed data, and using only night-time data in fact dramatically worsen the correlation with the simulation results.

Longer signals are not removed (band pass filter?). It would be a good idea to at least remove the trends before calculating correlations with ECCO. ECCO may have very different trends for different reasons. Furthermore, the use of climatological conductivity in elmgTD may falsify the trends of the ECCO EM results.

As previously explained, we do not use frequency methods because of the prevalence of gaps in the data. We believe that it is useful to retain all the signals remaining after our subtraction of tides and the linear trend.

Additionally, we do not believe using climatological conductivity data is a source of significant error. As shown in Grayver et al (2019), using an annual average seawater electric conductivity value versus using a monthly climatological value yields a difference of less than 0.005 nT in the induced oceanic electromagnetic signals for most of the globe (the plot of this is shown to the right). Thus, using the NOAA WOA seawater climatological values versus ECCO's seawater electric conductivity values is not likely to significantly alter our results.

[Figure]

3 - EM Prediction: Why are only the horizontal ECCO velocities used? Can the influence of the vertical velocities and their changes be quantified?

[Figure]

These figures show how the results vary when seafloor conductivity and vertical ocean flow are included. There is a great difference between using seafloor conductivity and omitting it. Meanwhile, including vertical ocean flow simply shifts the results by a small factor, and thus is less important for correlation studies. We will a discussion of this direct current shift in our future manuscript.

Line 110: What is meant by layer most closely corresponding to the sea floor? The models bottom layer? Do elmgTD and ECCO use a different bathymetry or does elmgTD bathymetry does not fit well to the reality? Maybe I just don't understand this sentence.

"To compare numerical predictions with the processed seafloor cable observations, the seafloor electric field was isolated by determining the electric field values of the depth layer most closely corresponding to the seafloor. These seafloor electric field values were then integrated along the path of a given cable, excluding the cable's continental endpoints. The results of this are shown and discussed in the next section."

We will revise that sentence and section so that the following information is clearer:
- The elmgTD solver uses the same vertical layers as ECCO (or whatever other ocean flow input model we use)
- ECCO uses 50 vertical layers at every grid point and each layer represents the same depth. However, the seafloor occurs at different depths across the ocean, so for each grid point the depth corresponding to the seafloor must be determined. This determined as the depth layer where the ocean velocities become zero.

4 - Results and discussion: In my opinion, some conclusions lack a solid base. To amend this, I would strongly recommend some recalculations or additional analyses. At the very least, the discussion should be deepened. The section's main arguments base on a mismatch between the ECCO results and the cable estimates. In short, two very different data sets are compared and if the observations do not fit to the model based estimates, then the observations are said to have a low "signal-to-noise-ratio".

Thank you for your comments. We agree that we should strengthen this discussion for the reasons you point out.

I would advise to repeat the elmgTD calculations with at least two other ocean models.

We investigated also using the GLORYS model. This model is higher resolution than ECCO and has significant differences in its velocities (see below figures).

[Figure]

Unsurprisingly, the predicted voltages varied dramatically between these two models, with no meaningful correlation between the models predicted cable voltages. These differences are largely due to differences in the top oceanic layer: the induced voltages depend on the depth-integrated velocities and the largest ocean currents are at the top layers. GLORYS is also a higher resolution, eddy-resolving model.

Thus, we have decided to continue by using the approach recommended by Reviewer #2 (see below): rather than try to compare the voltage predictions to one model and have a discussion that treats both the one model and the cable data as solidly reliable sources, we will discuss the limitations in this method and discuss how future cable studies may be better conducted by performing a principle component analysis (PCA) on synthetic data using the higher resolution GLORYS model. The goal of the PCA will be to determine the relationship between the induced cable voltage and the ocean flow's transport across the cable as it depends on both time and cable length.

The ECCO based results are not questioned or discussed at all. I have several questions here: How reliable are the ECCO results? What are the errors of the used velocities? There probably exist model inter comparison studies…Is the quality of the modeled data globally uniform? If not (probably not), then the "signal-to-noise-ration" mentioned in the paper depends not on cable length or the strength and uniformity of ocean currents but may just depend on location. ECCO is an assimilative ocean model: that means, that the errors of the results depend on the amount and quality of available data. This is not globally uniform, too. For example, if satellite altimetry is assimilated (major source of ECCO's information), then a big current like the Kuroshio (OKI cable) has a strong sea surface gradient and will be much better represented by the assimilated model than the more or less "flat" oceanic areas (HAW). But the HAW measurements might still be not worse than the OKI measurements. As long as it is not

clear if the found low correlations are caused by the model or the observations or the principal differences in the data, one should not call them low signal-to-noise ratio.

Balmaseda et al (2015) compares different ocean reanalyses. ECCO and GLORYS are both composed of data from satellites and in-situ measurements, and the assimilations are forced to also satisfy the laws of physics and thermodynamics. ECCO operates on a 1-degree global grid, whereas GLORYS uses a 0.25 degree grid and includes resolving eddies.

You bring up a very important point; ECCO uses a variety of satellite data to determine sea surface height and ocean currents. As you state, this better represents areas with stronger signals such as the Kuroshio Current and underrepresents "flatter" regions like the Eastern Pacific. The GLORYS model also has similar limitations. Indeed, any data-based model will.

We will revise the manuscript so it does not describe the cables as having a certain type of signal-to-noise ratio, but instead explicitly states the different sources of uncertainty entering the analysis from both the numerical work and the observational work.

The differences between modeled data and observed data are not discussed enough. Please discuss the effects in the cable data that are not in the modeled data: trends, ionosphere, solar cycle etc. see remarks to Sec. 2

We addressed this in your Section 2 comment and will be sure a similar discussion is also incorporated at this point of the paper.

Please discuss the representation error/issue: Grid box averages are compared to a very local cable path. By looking at Fig. 5C, one can see that even very similar cable paths can already produce very different results. Please discuss this.

Thank you for raising this point. In our updated simulations, along with calculating the results on a 1 degree grid, we also calculated them on transects of the cables' paths. This is the best way we can ensure error is not being introduced because of using numerical results from the wrong location. Of course, this method still is not perfect since there are no guarantees that the cables are perfect lines on the seafloor between their starting points.

In addition, from the differences between HAW1N and HAW1S some real signal-to-noise ratio could be derived. An error bar probably could be produced that sets the model to observation comparison into relation. Is there any explanation, why these two cables produce different time series (surely, the effects mentioned in the previous paragraph should affect both cables equally)?

Thank you for asking about this. The two cables are quite similar:

| HAW1N | | | | HAW1S | | |
|---|---|---|---|---|---|---|
| Range (V) | | Median (V) | \| mean \| (V) | Range (V) | | Median (V) | \| mean \| (V) |
| -27.845 | 15.7686 | -0.33576 | 1.9746 | -27.0786 | 16.6138 | 0.45492 | 2.0437 |

| (HAW1N - HAW1S) residuals |
|---|
| standard deviation (V) |
| 0.2117 |

We did a few more calculations on this since our analysis uses 90-day and 30.5-day knotted spline fits of the day—a process that in smoothing/averaging the data inherently lowers the observational error.

| HAW1N 90.5 day spline fit | | | | HAW1S 90.5 day spline fit | | |
|---|---|---|---|---|---|---|
| Range (V) | | Median (V) | \| mean \| (V) | Range (V) | | Median (V) | \| mean \| (V) |
| -0.3189 | 0.5203 | -0.0211 | 0.0271 | -0.3966 | 0.4989 | -0.0179 | 0.0255 |

| (HAW1N - HAW1S) 90.5 day spline fit residuals |
|---|
| standard deviation (V) |
| 0.0627 |

| HAW1N 30.5 day spline fit | | | | HAW1S 30.5 day spline fit | | | |
|---|---|---|---|---|---|---|---|
| Range (V) | | Median (V) | \| mean \| (V) | Range (V) | | Median (V) | \| mean \| (V) |
| -0.5158 | 1.0839 | 0.0033 | 0.0271 | -0.4974 | 0.6909 | 0.0174 | 0.0255 |

| (HAW1N - HAW1S) 30.5 day spline fit residuals |
|---|
| standard deviation (V) |
| 0.0816 |

Again, for the model side an error bar should be generated or estimated, too.

The dependence of the induced voltages on the flow velocities is linear on the global scale: i.e., increasing the flow velocities everywhere and at all times by 10% will give a 10% increase of predicted voltages. Although it does not hold locally in space or time, we can use this as a crude estimate of error: relative error of the predicted voltages is the same as the relative error of flows. There is no clear number of the estimated velocity error from the ECCO model; doing an intensive ensemble simulation could provide such an error estimate, however, this is beyond the scope of this study.

For the conductivity, the dependence is non-linear. On page 3, we show a figure illustrating the difference between using an annual average seawater electric conductivity value versus using a monthly climatological value.

Spline smoothed observations are compared to temporal averages from the model?

We'll add a sentence to the end of the Numerical Predictions section that explicitly states how the numerical results are compared to the simulations:

> For comparison to the spline smoothed observations, the numerical simulations' seafloor cable voltage predictions underwent the same 30.5-day knotted and 90-day knotted spline fits. These are shown in the results of Figure 5.

**Anonymous Referee #2**
The authors go through the commendable and accurate process of estimating oceanic electric fields from models, to compare with data from 4 submarine cables. The primary results presented are correlations between the observed and modelled electric fields, which are used to infer the suitability of using submarine cables for oceanic velocity. The statistical interpretation of these correlations does not seem methodical enough to be believable in its current state. The conclusions presented are not detailed, and do not advance the field beyond earlier papers on the topic. Even their recommendations for placing cables in strategic points - an easy thing to propose but much harder to actually implement, see the SMART cable effort - does not include the specificity needed to ensure that such cables can provide useful results

for inferring ocean circulation, such as resolving meanders, variables subsurface sediment thickness, or flow acceleration/deceleration. This article focuses on just the first step of getting useful cable voltage measurements, obtaining a high correlation between observations and models, but the second step of interpreting why the cable voltages change is just as important and even harder.

Thank you for your very thoughtful comments and review. We address your points below and very much appreciate your recommendations for bringing this study past the first step.

Technical comments:

*Intro*
lines 41-43: Another confounding factor is that, because longer cables integrate over longer distances, it becomes harder to assign transport or velocity to any single section of the cable.

Very true. We will adjust the sentence to become:
> …however, there are many challenges in using longer cables. These challenges are largely due to the myriad of processes which may also induce marine electromagnetic fields, especially across the length of the cable: secular variation (Shimizu et al., 1998), variations in ionospheric tides (Pedatella et al., 2012; Schnepf et al., 2018), geomagnetic storms or longer period ionospheric/magnetospheric signals (Lanzerotti et al., 1992a, 1995, 2001). Additionally, because the cable voltage is produced from the electric field integrated along the entire cable length, the longer the cable is, the more challenging it is to assign cross-cable ocean transport to any one section of the cable.

lines 44-45: This question has already been addressed in the literature.

Please let us know papers you are thinking of. We were not aware of any prior studies using data from cables spanning more than 1000km.

*Data and Data processing*
line 65: Also look at Luther publications from BEMPEX for an interpretation of the oceanic EF response at periods from hours to days.

Thank you for this reference recommendation for daily variation signals; we will include this reference in the revised manuscript.

*Section 3*
lines 86-93: Does elmgTD also include mildly conductive subsurface sediment layers, which vary significantly across ocean basins? These are important for interpreting oceanic EM signals.

elmgTD can include these subsurface sediment layers and our revised numerical work included them. They did significantly change the signal (see above figure on page 4).

Figure 3: What date/time are the ECCO velocities shown for?

Thank you for catching this. The caption will be revised accordingly:
> Figure 3. The surface velocities from ECCOv4r3 are shown in a) for the zonal (U) component and b) for the meridional (V) component. The labelled, thick black lines denote the seafloor voltage cables used in this study. A snapshot of the IGRF vertical main field, Bmainz is illustrated in c) and d) depicts the NOAA World Ocean Atlas seawater electrical conductivity's January climatology in

the surface layer. All snapshots represent conditions of January 17, 1997.

Figure 4: Why is this shown globally, when the focus is on the North Pacific? It would be more instructive to show the signed electric field across the cables, as the fact of taking the absolute magnitude hides the important fact of current reversals across the cables?

We included global plots so readers unfamiliar with this would be able to easily envision where cable studies may be more successful.

We appreciate your suggestion to make figures of the electric field across the cables. We will certainly include such figures in the next version of the manuscript. As we discuss below, we are undertaking further numerical work to perform the suggested PCA and such a figure will coincide very well with that analysis.

Figure 4: What time point do these figures apply to? The two pairs of plots (a and b; c and d) are slightly redundant: it doesn't matter what the surface or bottom fields/forcing are, but rather what their depth-integrated quantities are. Suggest reducing this figure to 2 plots showing the depth-averaged quantities.

This figure also applies to January 17, 1997 and the caption will be revised to explicitly state that.

We illustrate the sea surface and seafloor electric fields and forcing to help the reader understand that there is variation of the electric field with depth due to the significant vertical gradients of horizontal velocities, causing complex interactions between the poloidal and toroidal electric and magnetic fields (Velímský et al. 2019). These figures are meant to be informative in this way. We can try to condense these figures into something that more clearly shows this concept.

For now, here are additional figures that show the electric field at different depth layers across both time and colatitude. These figures are at a fixed longitudinal location and h represents the ocean depth. Left panels show the zonal component obtained using monthly conductivity climatology values; the right panels show the difference with respect to a run using a mean annual conductivity.

[Figure]

[Figure]

*Section 4, Results and Discussion*
line 119: I am highly surprised that "all of the p-values were equal to 0". Given that a p-value is positive definite by definition, I expect this to be impossible, certainly with real-world noisy data. This statistical significance testing needs to be redone more accurately. You need to also account for the degrees of freedom of a low-frequency signal.

We did not yet change this, but we appreciate your comment and will reform this section of the manuscript. We did some revised model runs to account for sediment conductivity and oceanic vertical flow, but are performing higher temporal resolution numerical simulations to conduct the PCA/EOF analysis you recommend below. We will incorporate improved correlation testing once that analysis is complete; in the version you reviewed, we were simply using Matlab's "corrcoef" function and it may be over simplifying our study by rounding low p-values down to zero.

For interpreting the HAW1N, HAW1S, and HAW3 voltages, it would be instructive to do statistical tests (see earlier comments) to see if the correlations of these cables with the model data are statistically distinguishable from each other. I am doubtful that they are.

It is unclear to us what you are suggesting here—are you suggesting testing of the numerical simulations for one cable have the same correlation with a given cable as those simulated for a different cable?

The additional simulations we did for the revision (i.e. using seafloor conductivity and vertical ocean flow), did not show correlation with the cables' data. We are still working on this update for the revised manuscript and will include it by early August.

lines 153-156: This is the crux of successfully using submarine cable voltages: placing it in a region that is conducive to interpreting such measurements. Note also that substantial effort is put into calibrating the Florida Current voltage time-series, see more recent publications by Meinen.

Thank you very much for this reference suggestion.

lines 157-163: Yes, most scientists who work with submarine cables could confirm that these are useful requirements for using such signals to interpret voltages. This point is not, however, substantiated in detail by this paper.

Nowhere do the authors note that their correlations are subject to an important additional source of noise: that the ECCO model might not accurately reflect the actual monthly averaged oceanic velocity field. To my knowledge nobody is able to evaluate ocean models based on their velocity field (for many practical and technical reasons). In light of this, a better approach, see Flosadottir et al 1997, would be to use a "perfect model" approach, so that you don't have to worry about the mismatch between ocean models and actual ocean circulation.

We are familiar with that paper and that certainly is an interesting approach. However, we feel that this first step of using actual (and very imperfect) seafloor cable data on a large (>1000 km) scale is an important aspect of our paper—even if the results only suggest more work is needed.

Instead of incorporating this approach, we are planning on incorporating the PCA/EOF approach you recommend below.

Also, for understanding the Florida Cable results, important details are presented in Spain and Sanford, J Mar Research, 1987.

Thank you very much for this reference recommendation.

To go along with my review above, what would actually be a very good contribution to understanding these 4 cables would be to delve into the oceanic flow that causes the voltages. Within the ECCO "perfect model" framework, how does variability and spatial structure of the oceanic flow field give rise to the cable voltages? This would provide a way to quantify the paper's unsubstantiated comments about the relative impact of coherent or incoherent flow. As cable voltage depends on the water transport above it, you could break the cables' voltage calculation from the ECCO model into shorter segments, and compare the multivariate segment-induced voltage against the transport over the segment. An EOF or Principal Component analysis seems best for explaining the voltages, and then coherence between the different segments can be calculated to quantify the degree of coherent cross-cable flow or not. Kudos if you look at variable segment lengths, especially calling out the spacing typical of telecom repeaters (see SMART cable material).

This is a very brilliant suggest—we are very grateful to you for putting it forward and we are thrilled to undertake this sort of analysis. Unfortunately, due to a variety of challenges these past few months, we have not yet completed the PCA, but we do anticipate including this in a revised manuscript by August 7, 2020.

Thank you again for your thorough and constructive review of our manuscript. We apologize for our slow response; these last few months have been very challenging for us, but things are started to settle int oa groove, so we look forward to wrapping up a revised version of the manuscript based on your suggestions in the coming month.

Sincerely,
The authors.

---

## Author Comment (AC2) · 30 Nov 2020

We sincerely thank the reviewer for these recommendations; however, we decided such work is beyond the scope of the current paper. Instead, in our Results and Discussion section, we acknowledge this:

"The qualitative comparison of the induced voltages and water transports along the cables, as presented in this paper, could be made more exact by applying the Principal Component Analysis/ Empirical Orthogonal Functions methodology. When applied only to the water transports provided by different ocean models, it could reduce the burden of calculating a detailed 3-D EM response to each model, and allow a more

focused interpretation of the observed voltages. We plan to carry out such analysis in the future."

---

## Author Comment (AC3) · 30 Nov 2020

**Anonymous Referee #2 (RC2)**
The authors go through the commendable and accurate process of estimating oceanic electric fields from models, to compare with data from 4 submarine cables. The primary results presented are correlations between the observed and modelled electric fields, which are used to infer the suitability of using submarine cables for oceanic velocity. The statistical interpretation of these correlations does not seem methodical enough to be believable in its current state. The conclusions presented are not detailed, and do not advance the field beyond earlier papers on the topic. Even their recommendations for placing cables in strategic points - an easy thing to propose but much harder to actually implement, see the SMART cable effort - does not include the specificity needed to ensure that such cables can provide useful results for inferring ocean circulation, such as resolving meanders, variables subsurface sediment thickness, or flow acceleration/deceleration. This article focuses on just the first step of getting useful cable voltage measurements, obtaining a high correlation between observations and models, but the second step of interpreting why the cable voltages change is just as important and even harder.

Thank you for your very thoughtful comments and review. We address your points below and very much appreciate your recommendations for bringing this study past the first step.

Technical comments:

*Intro*
lines 41-43: Another confounding factor is that, because longer cables integrate over longer distances, it becomes harder to assign transport or velocity to any single section of the cable.

Very true. We will adjust the sentence to become:
> …however, there are many challenges in using longer cables. These challenges are largely due to the myriad of processes which may also induce marine electromagnetic fields, especially across the length of the cable: secular variation (Shimizu et al., 1998), variations in ionospheric tides (Pedatella et al., 2012; Schnepf et al., 2018), geomagnetic storms or longer period ionospheric/magnetospheric signals (Lanzerotti et al., 1992a, 1995, 2001). Additionally, because the cable voltage is produced from the electric field integrated along the entire cable length, the longer the cable is, the more challenging it is to assign cross-cable ocean transport to any one section of the cable.

lines 44-45: This question has already been addressed in the literature.

Please let us know papers you are thinking of. We were not aware of any prior studies using data from cables spanning more than 1000km.

*Data and Data processing*
line 65: Also look at Luther publications from BEMPEX for an interpretation of the oceanic EF response at periods from hours to days.

Thank you for this reference recommendation for daily variation signals; we will include this reference in the revised manuscript.

*Section 3*
lines 86-93: Does elmgTD also include mildly conductive subsurface sediment layers, which vary significantly across ocean basins? These are important for interpreting oceanic EM signals.

elmgTD can include these subsurface sediment layers and our revised numerical work included them. They did significantly change the signal (see above figure on page 4).

Figure 3: What date/time are the ECCO velocities shown for?

Thank you for catching this. The caption will be revised accordingly:
> Figure 3. The surface velocities from ECCOv4r3 are shown in a) for the zonal (U) component and b) for the meridional (V) component. The labelled, thick black lines denote the seafloor voltage cables used in this study. A snapshot of the IGRF vertical main field, Bmainz is illustrated in c) and d) depicts the NOAA World Ocean Atlas seawater electrical conductivity's January climatology in the surface layer. All snapshots represent conditions of January 17, 1997.

Figure 4 comments

We have changed Figure 4 (and have also included similar Figures 5 and 6 for the HAW1NS and HAW3 cables).

> Figure 4. The results for the OKI cable. The top panel shows in red and green the smoothed time series of cable voltages using 30.5-day and 90-day knot separation, respectively. The blue and brown lines correspond respectively to the predictions obtained by the 3-D and 2-D model. The middle panel shows the time-development of voltage gradient along the cable length from the 3-D model. In the bottom panel, we plot in similar way the ECCOv4r4 vertically integrated transport across the cable. The cable orientation follows Table 1, from Honshu to Okinawa.

This new figure is shown on the next page.

[Figure]

*Section 4, Results and Discussion  comments*

We have substantially revised our Results and Discussion section. We also include an additional figure, Figure 7.

Figure 7. The cross-correlations function (CCF) between the observed and predicted voltages for individual cables.

[Figure]

In Figure 7, we calculated the cross-correlation functions (CCF) between the predicted and observed voltages using the 30.5-day knot separation datasets. Because of gaps present in the data, the Gaussian-kernel method (Rehfeld et al., 2011) was applied. All CCFs have their respective peaks at zero phase leg. The OKI, HAW3, HAW1N, and HAW1S signals show respective peak correlations of 0.48, 0.48, 0.23, and 0.04. It is obvious that the discrepancies between the predicted and observed voltages are 160 still large, and significant efforts are required both on the side of data processing and numerical modeling to reconcile the results.

On the side of numerical modelling, one could devise a comparison study between different ocean models. Indeed, we have used our model to predict the magnetic fields of the LSOMG model in the past (Velimský et al., 2019), and we have also attempted the calculation of the cable voltages for the eddy-resolving GLORYS ocean model (not shown here). One problem related to this approach is the volume of computational resources necessary to carry out the calculations. As the cable voltages are sensitive to local electric fields, the usual simplifications of the EM induction solver, based on the thin-sheet approximation, or representing the oceans by a single layer with integrated water transports and electrical conductances, are problematic (Šachl et al., 2019; Vel.mský et al., 2019). The single 5-year calculation of the full physical model presented here, with 50 ocean layers and spherical-harmonic truncation degree 240, required about 105 CPU-hours to

complete. Semi-global or regional modelling tools with local refinement ability are needed
for more accurate numerical studies.

lines 153-156: This is the crux of successfully using submarine cable voltages: placing it in a region that
is conducive to interpreting such measurements. Note also that substantial effort is put into calibrating the
Florida Current voltage time-series, see more recent publications by Meinen.

Thank you very much for this reference suggestion.

lines 157-163: Yes, most scientists who work with submarine cables could confirm that these are useful
requirements for using such signals to interpret voltages. This point is not, however, substantiated in
detail by this paper.

Nowhere do the authors note that their correlations are subject to an important additional source of noise:
that the ECCO model might not accurately reflect the actual monthly averaged oceanic velocity field. To
my knowledge nobody is able to evaluate ocean models based on their velocity field (for many practical
and technical reasons). In light of this, a better approach, see Flosadottir et al 1997, would be to use a
"perfect model" approach, so that you don't have to worry about the mismatch between ocean models and
actual ocean circulation.

We are familiar with that paper and that certainly is an interesting approach. However, we feel that this
first step of using actual (and very imperfect) seafloor cable data on a large (>1000 km) scale is an
important aspect of our paper—even if the results only suggest more work is needed.

Also, for understanding the Florida Cable results, important details are presented in Spain and Sanford, J
Mar Research, 1987.

Thank you very much for this reference recommendation.

---

## Author Response (AR3)

**Reviewer #1**

I have only minor recommendations and comments:

p3 l66: please add the tidal acronyms

Done.

p3 l77: I think this is not mentioned in Pedatella, 2012

Leaving "electromagnetic" from the main text as Pedatella (2012) discusses the seasonal variations of tides in general.

Fig.2: Maybe show one year only, the two top panels look very similar

We have considered it and decided against this recommendation. An additional information provided by this Figure as it is, is the level of sparsity and gaps in the HAW3 data-set.

Fig.4-6: Please unify the y-axis ranges of the top panel

Done.

Fig.7: Where is the negative lag? (does it go down as well?)

Negative time lag is unphysical (and yes, it goes down). Following recommendation by the Reviewer #2, we have reverted to a simple correlation table, and time lags are not discussed anymore.

**Reviewer #2**

I have 5 substantive comments:

(1) I urge the authors to explicitly state that ocean state estimates like ECCO do not represent with 100source of error include atmospheric and magnetospheric noise which is not modelled, and might not be fully removed from the data. This is why I recommended earlier a perfect model experiment, to avoid such issues. Their chosen approach to present their results does stand on its own, but now the correlations should serve as a lower bound.

Agreed and reformulated.

I am not aware of published results that analyze how global ocean models compare with direct velocity measurements - mostly because negative results are not published. One recent article does discuss this in passing, however. See Szuts et al., 2019.

Reference added. We agree, that differences between the ECCO velocity model and actual velocity field represent a substantial source of error.

With minor effort they could add more insight into their ECCO-based predictions: calculate the correlation between the estimated cable voltage and the net transport across the cable. With entirely numerical results, there are no sources of noise to decrease this correlation, and so such a correlation could be considered an upper bound to what is achievable with measurements.

This was a very useful suggestion and we have implemented it and extended the discussion.

(2) line 50: Be careful: before making such a bold claim in an interdisciplinary field a thourough literature search is required. I found a half dozen articles that already cover this topic, most of which cite publications listed in their bibliography. I'm also including a number of references that treat short submarine cables or related topics, for further background.

Corrected.

(3) The discussion would be more complete and thourough by adding a few sentences that discuss how the middle and bottom subplots of Figures 4, 5, and 6 agree or disagree.

Extended discussion added.

(4) For an instantaneous process like motional induction, lagged correlation analysis does not hae any useful interpretation to my knowledge, which suggests removing figure 7. What would be useful, however, would be to provide confidence limits on the correlations (at zero lag). This would give insight into how significant such comparisons are, though my previous comment about the low fidelity of ECCO circulation to the real ocean means that these correlations are lower bounds. In doing these statistics, don't forget that the time-series are red and thus their degrees of freedom are much reduced than the number of data points in the time-series. So, divide the duration of the time-series by the separation of the splines's knot separation (e.g. 30.5 days).

Due to inclusion of induction coupling with the mantle and ocean self-induction, the process is not completely instantaneous, but we agree that the importance of these effects is not crucial here, and lagged correlations were replaced by a simple table with zero-lag correlation coefficients. Due to gaps in the cable data, we have not implemented the error analysis of the correlation coefficients. As expected, a quick analysis using the jackknife method predicted unrealistically small error bounds, and we have decided against showing it.

(5) In terms of the correlation values, I find it hard to believe that the HAW1S and HAW1N time-series, which are very similar, have correlations as different as 0.23 and 0.04. Have you de-trending all three time-series prior to calculating correlations? Unremoved trends could easily account for the difference in their correlations.

Thank you for pointing to this oversight. In the previous version of the paper, only the mean value was removed. De-trending now significantly affects the correlations, and also the presentation of voltages in the top panels of Figures 4–6. Best correlations are now obtained for HAW3 cable, and the difference between the N and S branches of the HAW1 cable are reduced.

Detailed comments:

line 6: "numerical predictions of the electric field induced by ocean circulation" doesn't make clear that the ocean circulation itself is an estimate.

Corrected.

line 7-8: "correlation between cable voltage data and numerical predictions strongly depends on both the strength and coherence of the velocities flowing across the cable," surely the correlation also depends on whether the numerical models accurately reproduce the true ocean signals (velocity and the resulting EM fields) sampled by the cable? Unless you can prove that the ocean models accurately predict ocean velocities - and there are far too few velocity observations to make such an estimate - then lack of correlation between cables voltages and numerical predictions is merely a representation issue that prevents answered the research question posed.

Reformulated.

line 19-20: To my knowledge, Faraday's experiment was inconclusive and "unsuccessful"; "not very successful" suggests some level of success.

Corrected.

Figures 4, 5, and 6: These color contour plots really make my eyes hurt. Especially for the sake of color-blind individuals and black-and-white printouts, please use a simple 2-toned colormap like in Figure 3 instead. I find I can extend the dynamic range of colors by transitioning from red/blue to black at the end of the positive/negative range, which does not hinder the ability to visually separate positive regions from negative regions. See Light and Bartlein (2004), and updates to this (e.g. https://betterfigures.org/2014/11/18/end-of-the-rainbow/ )

Color scales changed.

Figures 4, 5, and 6: How do you calculate transport? In oceanography, transport is volume per time, or m3/s. The legend for the bottom subplot says m2/s, however, which is transport per unit width. Please make consistent.

By $T_\perp$ we mean the vertically integrated horizontal velocities, perpendicular to the cable, thus the unit $\mathrm{m}^2/\mathrm{s}$. In the extended correlation analysis, we now use also the total transport across the cable $P_\perp$ in $\mathrm{m}^3/\mathrm{s}$.

Figure 4: Can you explain the relation between the voltage and transport plots in a little more detail? why there is strong transport from 250 km to 600 km, but instead the voltage induced in the cable is only positive around 150-250 km and 250-540 km? Why is the negative transport from 700-1200 km not reciprocated in voltage? Why does the weak positive transport from 0-150 km have a negative induced voltage?

We assign these differences to the complicated structure of electrical conductivity and ocean velocity in the vicinity of the Ryukyu arc. Large bathymetry changes have strong influence on the local EM fields.

Figure 4: For the top subplot, the caption refers to a brown line, but none is visible.

A reference to a run with simplified 2-D EM physics, that we decided not to show. Removed.

Figures 5 and 6: There is closer correspondence between the transport and voltage in these two examples compared with Figure 4. I would be interested to know why.

Because we are in the deep ocean with smaller variations in bathymetry and no resistive continents nearby.

Figure 7: Unless you discuss the lagged correlations, this figure is unnecessary.

Done. Replaced by Table 2.